# Genome-Wide Identification and Characterization of the *CCT* Gene Family in Foxtail Millet (*Setaria italica*) Response to Diurnal Rhythm and Abiotic Stress

**DOI:** 10.3390/genes13101829

**Published:** 2022-10-10

**Authors:** Yuntong Li, Shumin Yu, Qiyuan Zhang, Ziwei Wang, Meiling Liu, Ao Zhang, Xiaomei Dong, Jinjuan Fan, Yanshu Zhu, Yanye Ruan, Cong Li

**Affiliations:** College of Bioscience and Biotechnology, Shenyang Agricultural University, Shenyang 110866, China

**Keywords:** *Setaria italica*, diurnal rhythm regulation, *CCT*, hormones, abiotic stress

## Abstract

The *CCT* gene family plays important roles in diurnal rhythm and abiotic stress response, affecting crop growth and development, and thus yield. However, little information is available on the *CCT* family in foxtail millet (*Setaria italica*). In the present study, we identified 37 putative *SiCCT* genes from the foxtail millet genome. A phylogenetic tree was constructed from the predicted full-length SiCCT amino acid sequences, together with CCT proteins from rice and *Arabidopsis* as representatives of monocotyledonous and dicotyledonous plants, respectively. Based on the conserved structure and phylogenetic relationships, 13, 5, and 19 SiCCT proteins were classified in the *COL*, *PRR*, and *CMF* subfamilies, respectively. The gene structure and protein conserved motifs analysis exhibited highly similar compositions within the same subfamily. Whole-genome duplication analysis indicated that segmental duplication events played an important role in the expansion of the *CCT* gene family in foxtail millet. Analysis of transcriptome data showed that 16 *SiCCT* genes had significant diurnal rhythm oscillations. Under abiotic stress and exogenous hormonal treatment, the expression of many *CMF* subfamily genes was significantly changed. Especially after drought treatment, the expression of *CMF* subfamily genes except *SiCCT32* was significantly up-regulated. This work provides valuable information for further study of the molecular mechanism of diurnal rhythm regulation, abiotic stress responses, and the identification of candidate genes for foxtail millet molecular breeding.

## 1. Introduction

Many environment factors, including abiotic stresses and photoperiodic rhythm, affect growth and development, and thus yield [1,2,3]. The molecular mechanisms by which plants adapt to environmental changes are very complex, and a number of gene families are involved in the regulatory pathways. The *CCT* gene family has been shown to play important roles in plant response to environmental changes, including circadian rhythm and abiotic stress [4,5,6]. Structural analysis and functional study of *CCT* family genes in plants will help us to explore the molecular mechanism of environment stress response, which is significant for improving the environmental adaption of plants.

The *CCT* family genes contain a conserved CCT domain and are primarily known in flowering plants [7]. According to the distribution of conserved domains, *CCT* family genes are classified into three subfamilies, comprising the CCT motif (CMF) subfamily that contains a single CCT motif [8], the CONSTANS-like (COL) subfamily that contains a CCT domain and one or two zinc-finger B-Box domains [9], and the PSEUDORESPONSE REGULATOR subfamily (PRR) that possesses a pseudo-receiver domain and a CCT motif [5]. The study of cormophyte and streptophyte lineages indicates that the *CMF* subfamily evolved from the *COL* subfamily through the loss of the B-Box domains [8]. These changes in the domains further enhanced the functional diversification of the *CCT* family.

To date, many members of the *CCT* gene family have been identified and extensively studied in certain plant species, including *Arabidopsis* (*Arabidopsis thaliana*), rice (*Oryza sativa*), maize (*Zea mays*), and wheat (*Triticum aestivum* L.) [4,10,11,12]. The first *CCT* family gene was cloned from *Arabidopsis* and was named *CONSTANS* (*CO*), which controls the flowering time of Arabidopsis [4]. The *CO* gene regulates the expression of *FLOWERING LOCUS T* (*FT*), which encodes a mobile systemic signal, to promote flowering in *Arabidopsis* by binding to the *cis*-acting elements TCTC (N2–3) ATG in the promoter of *FT* and to accelerate plant flowering under long days (LD) [13]. *HEADING DATE 1* (*HD1*), which is homologous with *CO* and was cloned from rice by map-based cloning, promotes heading under short days (SD) and suppresses heading under LD [11]. *HD1* regulates *HD3A* and *RICE FLOWERING LOCUS T 1* (*RFT1*), highly homologous *FT*-like genes of rice, and further regulates rice heading [14,15]. Some members of the *CCT* family can regulate the growth and development of plants under different photoperiods [10,11,15]. The circadian clock is a central part of these photoperiod regulation processes, and members of the *PRR* subfamily members are key components of the circadian clock regulatory network [5]. For example, five *PRR* genes (*PRR9*, *PRR7*, *PRR5*, *PRR3*, and *TIMING OF CAB EXPRESSION1* (*TOC1*)) in *Arabidopsis* are regulated by the circadian clock, and the expression levels of them show diurnal rhythm oscillations with light/dark cycles [16]. *PRR9*, *PRR7,* and *PRR5* can regulate the circadian clock by regulating *TOC1* expression, while *TOC1* affect the expression of *CIRCADIAN CLOCK ASSOCIATED1 (CCA1)* and *LATE ELONGATED HYPOCOTYL (LHY)* [7,17,18]. 

In addition to regulating flowering time, members of the *CCT* family perform several other functions. In *Arabidopsis*, *TOC1* regulates ABA signal relative genes (*ABAR/CHLH*) to be involved in drought tolerance [19]. *PRR7* regulates stomatal conductance and participates in oxidative stress response [20]. *Arabidopsis CONSTANS-LIKE 4* (*AtCOL4*) is strongly induced by abscisic acid (ABA) and osmotic stress, and *AtCOL4*-overexpressing plants show enhanced salt stress tolerance compared with the wild type [21]. As a negative regulator, *HEADING DATE 7* (*GHD7*), a member of the *CMF* subfamily member in rice, is involved in its growth, development, and regulation of abiotic stress processes. The knock-down of *GHD7* enhances drought tolerance [22]. In rice, *GHD2*, a *COL* subfamily member, performs similar roles as *GHD7* and interacts with *OsARID3* and *14-3-3* genes to participate in the regulation of drought stress responses and leaf senescence. The knockout mutant of *GHD2* has significantly enhanced drought tolerance, whereas overexpression of *GHD2* increases sensitivity to drought stress [23]. In addition to *Arabidopsis* and rice, *CCT* genes with similar functions are reported in maize. The maize *ZmCCT* gene delays flowering time and enhances drought tolerance by repressing the expression of *Vascular Plant One Zinc Finger 1* (*ZmVOZ1*) and *Arabidopsis Response Regulator* 16 (*ZmARR16*) [24]. Therefore, *CCT* family members play a particular role in plant regulating diurnal rhythm, growth and development, and resistance to abiotic stress. 

Foxtail millet was domesticated and cultivated in arid and semi-arid regions 8,700 years ago [25]. Foxtail millet is an important food crop in northwestern China that shows strong drought tolerance and sensitivity to change in photoperiod [26]. In addition, it has the characteristics of a small genome, short life cycle, and abiotic stress tolerance [27,28]. The study of the function of *CCT* family genes in foxtail millet is helpful for the domestication and breeding of foxtail millet. In this study, we identified 37 putative *SiCCT* genes and classified the genes into three subfamilies. We conducted a comprehensive bioinformatic analysis of the gene location, chromosomal distribution, exon–intron structure, motif composition, regulatory sequences, phylogenetic relationships, and synteny. The expression of the *CCT* genes was analyzed in different tissues by semiquantitative RT-PCR; diurnal rhythm oscillations were investigated using transcriptome data, and expression levels under different treatments were assessed by quantitative real-time PCR (qRT-PCR) analysis. The results provide insights into the function and evolution of the *SiCCT* gene family and its potential roles in foxtail millet growth and stress responses and provide a foundation for molecular breeding to improve the stress tolerance of foxtail millet.

## 2. Materials and Methods

### 2.1. Gene Identification, Chromosomal Location, and Phylogenetic Relationships of CCT Family Members in Foxtail Millet

The genome sequence of millet was obtained from the Ensembl Plant database (http://plants.ensembl.org/index.html, accessed on 30 September 2021). We downloaded the hidden Markov model of the CCT domain from the Pfam database (http://pfam.sanger.ac.uk/, accessed on 30 September 2021). The *SiCCT* family members were extracted from the genome database with HMMER 3.1 (http://hmmer.janelia.org, accessed on 30 September 2021). The candidate genes were screened against the National Center for Biotechnology databases (https://www.ncbi.nlm.nih.gov/cdd/, accessed on 12 October 2021) to determine the 37 final genes. Information on the CCT proteins were calculated using the ExPaSy ProtParam tool (http://web.expasy.org/protparam/, accessed on 12 October 2021), including length, molecular weight, and isoelectric point. We used the subcellular localization prediction tool Plant-mPLoc (http://www.csbio.sjtu.edu.cn/bioinf/plant-multi/, accessed on 12 October 2021) [29] to predict the likely location of the protein. The MapChart software [30] was used to map the chromosomal locations of the genes.

The protein sequences of the CCT domain-containing genes of Arabidopsis and rice were downloaded from the Ensembl Plant database on the basis of relevant research reports [5,8,9,31,32]. The full-length CCT sequences of *Arabidopsis*, rice, and foxtail millet were selected for comparison with MEGA X version 7 software [33]. Subsequently, a multiple sequence alignment was used to construct a maximum likelihood phylogenetic tree with the MEGA software with the following parameters: Poisson model, pairwise deletion, and 1000 bootstrap replications.

### 2.2. Collinearity Analysis and Gene Duplication

To explore the synteny of orthologous *CCT* genes obtained from foxtail millet and other selected species, we used the MCScan (Python version) tool (https://github.com/tanghaibao/jcvi/wiki/MCscan-(Python-version), accessed on 25 October 2021) to draw the synteny map. When visualizing the results, the gene filtering parameter in the small collinearity block was set to 30. Circos [34] was used to map all *SiCCT* genes to foxtail millet chromosomes based on physical location information from the millet genome database. A multiple collinearity scanning kit (MCScanX) [35] was used to analyze gene duplication events. KaKs_Calculator 2.0 [36] was used to calculate *K*_a_ and *K*_s_ to detect replication events. 

### 2.3. Gene Structure and Conserved Sequence Analysis

The location of introns, exons, and untranslated regions in the genes was extracted from the gene finding format (GFF3) file. Conserved domains of millet CCT proteins were elucidated with the MEME Suite (http://meme-suite.org/, accessed on 1 November 2021) [37]. The optimized parameters were employed as follows: the maximum number of motifs was 10, and the optimum width was from 6 to 50. The map of the gene structure and motifs was drawn using TBtools [38]. 

### 2.4. Cis-Acting Element Analysis

The genomic DNA sequence 1000 bp upstream of the translation initiation codon for the *SiCCT* genes was used for *cis*-acting element analysis. The PlantCARE database (http://bioinformatics.psb.ugent.be/webtools/plantcare/html/, accessed on 7 November 2021) [39] was used to predicte the promoter elements in the sequences. 

### 2.5. Plant Materials and Treatments

Millet ‘Yugu1’ was used in this study. Plants were grown in fields and in a greenhouse in Shenyang, Liaoning Province, China. The stress-treated plants were grown in a greenhouse in Hoagland’s nutrient solution [40]. Abscisic acid, salt stress, and drought stress were applied separately by treating plants with 100 μM ABA, 200 mM NaCl solution, or 20% PEG6000, respectively. Plant leaves were collected after treatment for 0, 4, 8, and 12 h, with three biological replicates for each treatment. For low-temperature treatment, plants were placed in a 4 °C incubator for 12 h, and samples were collected at 0, 6, and 12 h. The leaf, immature seed, spikelet, root, leaf sheath, shoot apical meristem, stem, im-mature leaf, and seed were collected separately at three developmental stages from field-grown foxtail millet plants for semiquantitative RT-PCR.

### 2.6. Analysis of CCT Gene Expression in Millet and Quantitative Real-Time PCR

Total RNA was extracted from each tissue using a plant RNA extraction kit (Accurate Biotechnology Co., Ltd., Changsha, China) in accordance with the manufacturer’s instructions. The cDNA was synthesized from 2 µg total RNA with M-MLV reverse transcriptase (Accurate Biotechnology Co., Ltd.). Semiquantitative PCR was performed using GoTaq^®^ Green Master Mix (Promega, Shanghai, China) with specific primers (synthesized by Genewiz, Suzhou, China). The GoTaq^®^ qPCR Master Mix kit (Promega) was used for qRT-PCR reactions using a C1000 Thermal Cycler and quantified using a CFX96 Real-Time System (Bio-Rad, Hercules, CA, USA). The relative transcript level was calculated using the 2^−ΔΔ*C*t^ method [41]. Primers used in this experiment are enlisted in Appendix A. Actin gene (*SETIT_010361mg*) was used as internal control for this experiment. Transcriptome data used in this study were downloaded from the NCBI SRA database under accession number PRJNA684771 [42]. We processed per kilobase per million (FPKM) data for diurnal rhythm analysis. Yi et al. [42] sampled the two groups of samples at the beginning of light and the beginning of darkness, respectively, and there were two groups of data at 16 h, 20 h, and 24 h. Therefore, we draw the line graph using the average of the FPKM values of these two sets of data.

## 3. Results

### 3.1. Identification and Analysis of CCT Gene Family Members in Foxtail Millet

A total of 37 predicted *CCT* genes were extracted from the foxtail millet genome using a hidden Markov model (PF06203). The genes were designated as genes from *SiCCT1* to *SiCCT37* based on their chromosomal distribution (Table 1). The gene identifier, genomic position, predicted molecular weight, protein length, and isoelectric point of each gene are listed in Table 1. The length of the proteins ranged from 160 aa (*SiCCT27*) to 760 aa (*SiCCT35*); the molecular weight varied from 17.6 kDa (*SiCCT27*) to 82.7 kDa (*SiCCT35*), and the isoelectric point ranged from 4.37 (*SiCCT32*) to 10.18 (*SiCCT27*). All of the genes were predicted to be localized in the nucleus except *SiCCT4*. *SiCCT4* was predicted to be localized in both the mitochondria and the nucleus.

Thirty-six *SiCCT* genes were mapped to eight of the nine foxtail millet chromosomes. The number of *SiCCT* genes varied among the chromosomes, and no genes were detected on chromosome V (Figure 1). Four chromosomes (IV, VI, VII, and VIII) contained three genes, and two chromosomes (II and III) carried four genes. Chromosome IX contained the largest number of nine genes. One gene was not mapped to a chromosome and was recategorized as unplaced (Figure 1). *SiCCT14* and *SiCCT15* were located on chromosome 2 and identified as a pair of tandem repeats.

### 3.2. Evolutionary and Synteny Analysis of CCT Genes in Foxtail Millet and Other Species

To detect the evolutionary relationships among *CCT* family members, a maximum likelihood phylogenetic tree was constructed based on 99 full-length CCT protein sequences, comprising 32 from *Arabidopsis*, 30 from rice, and 37 from foxtail millet (Figure 2 and Appendix A). The CCT proteins were resolved into three main clades consisting of eight groups, which were designated A to H. Group B represented the *PRR* subfamily with 15 members. The *CMF* subfamily comprised groups A (23 members), C (4 members), E (5 members), and G (9 members). The remaining groups (D with 16 members, F with 18 members, and H with 9 members) belonged to the *COL* subfamily. 

To enable further inference on the phylogenetic affinities of the foxtail millet *CCT* family, synteny analyses were performed between the *CCT* genes of foxtail millet and other species, comprising the dicotyledons *Arabidopsis*, tomato (*Solanum lycopersicum*), and alfalfa (*Medicago truncatula*) and the monocotyledons rice, sorghum (*Sorghum bicolor*), and maize (Figure 3). Only one pair of homologous genes between foxtail millet and *Arabidopsis*, and two pairs between foxtail millet and each of alfalfa and tomato, were detected. A greater number of gene pairs were detected between foxtail millet and the other monocotyledonous species, namely rice (26 orthologous gene pairs), sorghum (28 orthologous gene pairs), and maize (31 orthologous gene pairs). Among these genes, several genes comprised more than three pairs in a collinear relationship between foxtail millet and the other species (Appendix A), such as for *SiCCT34*, *SiCCT35*, and *SiCCT36*. *SiCCT34* was detected in all studied species except *Arabidopsis*, suggesting that this gene may be evolutionarily conserved and may have played an important role in the evolution of *CCT* gene family.

Collinearity within the *SiCCT* family was analyzed to explore segmental duplication events within the *SiCCT* genes. Six pairs of segmental-duplicated genes were found on foxtail millet chromosomes (Figure 4). We calculated the nonsynonymous substitution rate (*K_a_*)/synonymous substitution rate (*K*_s_) ratio for each gene pair to examine the evolutionary mechanism of the *CCT* genes (Table 2). All *K*_a_/*K*_s_ values were less than one, except for *SiCCT32*/*SiCCT36*. To further explore the evolution of *CCT* genes in other Poaceae species, we also selected the *CCT* genes of maize and sorghum to plot the respective intraspecific collinearity relationships (Appendix A and Appendix A). Twenty-three and 10 pairs of segmental-duplicated genes were found in the maize and sorghum genomes, respectively.

### 3.3. Gene Structure and Conserved Motif Analysis

The phylogenetic tree of *SiCCTs* was constructed and divided into 10 groups (I to X; Figure 5A). To better understand the evolution of *CCT* genes in foxtail millet, we analyzed the intron–exon structure of the identified *CCT* genes (Figure 5B). Most *SiCCT* genes contained one to three introns (eight genes with one intron, seven with two introns, and seven with three introns). A smaller proportion of the genes contained four to eight introns (two genes with four introns, four with five introns, three with six introns, four with seven introns, and one with eight introns). Only *SiCCT20* lacked an intron. 

We constructed domain diagrams of the 37 predicted *SiCCT* genes to study their conserved motifs using the MEME Suite of sequence analysis tools (Figure 5C). The CCT domain was composed of motifs 1 and 3, and all *SiCCT* genes had complete CCT institutional domains, except for *SiCCT14*, *SiCCT15*, and *SiCCT24*. *SiCCT14* only had motif 3, and *SiCCT24* and *SiCCT15* only had motif 1. Motif 2 was the B-Box domain, which was specific to the *COL* family. The PRR domain was composed of motif 5 and motif 10. In the PRR subfamily, *SiCCT9*, *SiCCT4*, *SiCCT35*, and *SiCCT11* had intact PRR domains, whereas *SiCCT26* only had motif 10. Furthermore, we found that *SiCCT29*, *SiCCT31*, *SiCCT27,* and *SiCCT18* had independent evolutionary branches in the phylogenetic tree and their CCT domains (motif 1 and 3) were closer to the 5′ end than other *CCT* genes. These results indicated that the domains of these proteins were relatively conserved and conformed with the subfamilial classification of the *CCT* family. 

### 3.4. Analysis of Cis-Acting Elements in the SiCCT Gene Promoter Region

To predict the function and regulatory mechanism of the *SiCCT* genes, we analyzed the *cis*-acting elements in the promoter region of the genes. Based on the results of a PlantCARE analysis, the *cis*-acting elements were divided into three categories: plant growth and development, abiotic and biotic stresses, and phytohormone responsive (Appendix A). We selected 27 *cis*-acting elements from the promoter of the *SiCCT* genes and divided them into the three categories (Figure 6 and Appendix A). In the growth and development group, the G-Box element involved in light response was most numerous and was detected in 24 of the 37 genes. In addition to light-response elements, the CAT-Box element involved in the regulation of meristem expression was present in 14 genes. An O_2_-site element involved in zein metabolism regulation was detected in nine genes. In the abiotic and biotic stresses group, multiple response elements were observed, such as oxidation, (ARE and GC-motif), wounding (box-S and WUN-motif), drought (MBS), high temperature (STRE), low temperature (LRE), and defense (CCAAT-box). The most numerous elements in the stress-response group were MYB and MYC binding sites, which are general stress-responsive elements. In the phytohormone-responsive group, we detected many elements involved in hormone response, including elements responsive to ABA (ABRE), salicylic acid (as-1), indoleacetic acid (AuxRR-core and TGA-element), gibberellins (GARE-motif and P-box), and methyl jasmonate (TGACG-motif). Among them, the number of ABRE elements was 92, which was the largest number of elements related to hormone regulation.

### 3.5. Expression Analysis of SiCCT Genes in Different Tissues and under Abiotic Stress and Exogenous Hormone Treatments

We used semiquantitative RT-PCR to investigate the expression of the *SiCCT* genes in various tissues, comprising the leaf, immature seed, spikelet, root, leaf sheath, shoot apical meristem, stem, immature leaf, and seed (Appendix A). Four genes (*SiCCT17*, *SiCCT22*, *SiCCT26*, and *SiCCT31*) were highly expressed; three genes (*SiCCT18*, *SiCCT24*, and *SiCCT37*) were weakly expressed, and two genes (*SiCCT14* and *SiCCT16*) were not expressed in all tissues. The RT-PCR results showed that the *SiCCT* genes were expressed in various tissues and differed in expression patterns. 

In the cis-element analysis, a large number of elements related to light response and diurnal rhythm were detected in the promoter region of *CCT* genes. To further explore the *CCT* genes response to diurnal rhythm, an expression analysis of *CCT* genes under diurnal rhythm was performed. The transcriptome data (PRJNA684771) used for diurnal rhythm analysis were downloaded from NCBI database according to the research of Yi et al. [42]. The expression pattern of 16/37 *CCT* genes shoedn diurnal rhythm oscillations, which includes all *PRR* members, 6 *COL* members, and 5 *CMF* members (Figure 7 and Appendix A). The expression of *SiCCT4* and *SiCCT9* reached the highest level at 12 h, and the expression of *SiCCT11* and *SiCCT35* reached the highest at 8 h, while *SiCCT26* showed different rhythm oscillation than other *PRR* members. In addition to the *PRR* family genes, the *COL* subfamily members (*SiCCT2*, *5*, *7*, *8*, *16*, and *17*) and *CMF* subfamily members (*SiCCT19*, *28*, *31*, *32*, and *33*) also showed diurnal rhythms, and the expression peak of these genes appeared in different times. Among them, the expression peak of *SiCCT5*, *8*, *16*, *19*, and *31* reached the highest level at dark.

Many genes in the *CMF* subfamily have been reported to be involved in abiotic stress and hormone regulation in plants [22,24]. To clarify whether foxtail millet *CMF* subfamily genes responded to abiotic stresses and ABA treatment, we analyzed their expression patterns under different treatments (Figure 8). Overall, the transcript abundance of many genes was significantly increased, especially under ABA and drought treatment. After ABA treatment, all genes were up-regulated except *SiCCT3* and *SiCCT12*. Among the up-regulated genes, the expression of *SiCCT13* was the highest, significantly up-regulated by more than 29 times, and its expression peaked at 8 h. Under drought induction, the expression levels of all genes were induced except *SiCCT22*, *SiCCT29*, and *SiCCT32*. The expression level of *SiCCT3* was significantly up-regulated by 40-fold at 12 h. The expression levels of half of the genes were induced by salt treatment, and the expression of most genes was induced by more than three times. Eight genes responded to low-temperature treatment, of which the expression levels of five genes were induced and three genes were inhibited. *SiCCT31* was most significantly induced by low temperature, and the expression level of *SiCCT31* was up-regulated by more than eight times at 8 h.

In general, most of the *CMF* subfamily genes showed a trend for up-regulated expression in response to abiotic stress and hormone treatment (Appendix A). After ABA treatment, 10 genes were up-regulated, followed by drought treatment. The expression of *SiCCT37* was up-regulated under the four treatments. The expressions of *SiCCT21*, *SiCCT28*, and *SiCCT18* was induced by exogenous ABA, salt, and drought. The number of genes down-regulated by salt treatment were the most among all treatments (Appendix A). The expressions of *SiCCT22* and *SiCCT29* was inhibited by salt and drought.

## 4. Discussion

Stress tolerance and photoperiod regulation are important factors for plant adaption to growth in different regions. The *CCT* gene family has been reported to be involved in the regulation of photoperiod and stress responses in plants [13,43,44]. *CCT* genes have been identified in many crop species at the whole-genome level. For example, 41 and 53 *CCT* family genes have been identified in rice [45] and maize [44], respectively. Foxtail millet exhibits the characteristics of strong tolerance to abiotic stress and sensitivity to photoperiod. Therefore, we identified possible *CCT* genes from the foxtail millet genome. In this study, 37 putative *CCT* family genes were identified in foxtail millet (Table 1). A phylogenetic analysis of the *CCT* family genes of foxtail millet, *Arabidopsis*, and rice resolved the genes into eight groups within three clades (Figure 2). Several pairs of *SiCCT* genes homologous to the reported *OsCCT* and *AtCCT* genes were found in the phylogenetic tree. For example, *SiCCT6* was homologous to *HD1* and *CO*, *SiCCT5* and *SiCCT8* were homologous to *DTH2*, *SiCCT35* was homologs to *HD2*, and *SiCCT4* and *SiCCT26* were homologous to *APRR1*. The high homologies among the genes may imply that the proteins perform similar functions. 

Whole-genome duplication (WGD) and nested chromosome fusions (NCFs) are important driving forces in the evolution of flowering plants [46,47]. WGD can provide a plant with the opportunity for diversification in gene functions, and NCFs can change the number of chromosomes. These processes are important for speciation and the evolution of novel gene functions. Flowering plants have experienced many WGD events [48], of which three WGD events have occurred in the genomes of cereal grasses [49]. To explore the evolution of *SiCCT* genes in cereals, we constructed a collinearity map for maize and sorghum (Appendix A and Appendix A) and calculated the *K*_a_ and *K*_s_ values for foxtail millet, sorghum, and maize (Table 2). We then estimated the timing of the doubling event of the *SiCCT* genes. The doubling time points for sorghum and maize were also calculated as controls. Foxtail millet separated from maize and sorghum approximately 27 million years ago (Mya), and maize and sorghum diverged approximately 13 Mya [25]. Based on the estimated dates for these two nodes and the respective estimated divergence dates for the Poaceae (98.2 Mya) [50] and the subfamily Panicoideae (48 Mya) [25], we divided the calculated WGD events into time periods comprising an intermediate ancestor period (~98.2 Mya), Poaceae ancestor period (~98.2 Mya to ~48 Mya), Panicoideae ancestor period (foxtail millet: ~48 Mya to ~27 Mya; maize and sorghum: ~48 Mya to ~13 Mya), and post-speciation period (foxtail millet: ~27 Mya; maize and sorghum: 13 Mya). Based on the calculations, the *CCT* genes of foxtail millet and sorghum diverged in relatively ancient times, with the diversification of most of the genes concentrated between the intermediate ancestor period and the Poaceae ancestor period. These findings suggest that *SiCCT* genes have hardly changed after speciation. From an evolutionary point of view, *SiCCT* genes have been essentially stable since entering the Panicoideae ancestor period. These results indicate that the functions of *SiCCT* genes were determined before the divergence of foxtail millet, which further implies that the functions of the genes are conserved. 

Tandem repeats and fragment repeats gave rise to 14 genes, which were important for *CCT* family amplification in foxtail millet (Figure 1 and Figure 4). We speculate that the generation of these new genes might be caused by changes in the domains. The generation of the *CMF* subfamily resulted from the loss of the B-Box domain of the *COL* subfamily members [5]. Therefore, we investigated the domains of SiCCT proteins and observed that most proteins had complete CCT domains except SiCCT14, SiCCT15, and SiCCT24 (Figure 5). The CCT domain was composed of motifs 1 and 3. SiCCT15 and SiCCT14 had no other domain differences except in the CCT domain (SiCCT15 only had motif 1 and SiCCT14 only had motif 3). Both SiCCT14 and SiCCT15 were tandem repeats. Therefore, we consider that SiCCT14 and SiCCT15 were generated during the process of tandem duplication, owing to the separation of motif 1 and motif 3 in the CCT domain. The differences of SiCCT14 and SiCCT15 were not detected in SiCCT24. However, *SiCCT24* formed a collinear pair with *SiCCT25*. The motif 3 in SiCCT24 may have been lost when segment duplication caused by the WGD occurred. In addition, the protein structure of six pairs of collinear genes were compared. Three pairs were observed to be structurally different (SiCCT24/SiCCT25, SiCCT32/SiCCT36, and SiCCT2/SiCCT17). In addition to the loss of motif 3 in SiCCT25 to form SiCCT24, SiCCT36 lost three domains (motifs 4, 7, and 9), and SiCCT17 lost the B-Box (motif 2) to form SiCCT32 and SiCCT2, respectively, during the WGD event. Previous studies have shown that the deletion of the B-Box domain affects grain vernalization and response to LD [8]. Therefore, after domain deletion, these proteins are likely to have assumed novel functions, which are worthy of further exploration. Furthermore, we found that *SiCCT29*, *SiCCT31*, *SiCCT27,* and *SiCCT18* had independent evolutionary branches in the phylogenetic tree and their CCT domains were closer to the C terminus of protein than other *CCT* genes. This suggested that they may had different functions with other *SiCCT* genes.

It had been found that *CCT* genes responded to photoperiod signal and are involved in diurnal rhythm regulation, therefore affecting the flowering time of plants [44,51]. A number of elements related to light response and diurnal rhythm regulation were found on the promoter of *SiCCT*, including G-BOX, I-box, Sp1, and so on. Therefore, we further analyzed the expression rhythmicity of the *CCT* gene according to the transcriptome data. *PRR* subfamily genes are key factors in the regulation of plant diurnal rhythm. Figure 7 showed that the expression of *SiCCT4*, *SiCCT9*, *SiCCT11*, and *SiCCT35* had strong diurnal rhythm. Among them, the expression of *SiCCT11* and *SiCCT35* reached the highest value at 8 h of light, and they had similar expression patterns in diurnal rhythm. In addition, the phylogenetic tree analysis showed that both *SiCCT35* and *SiCCT11* were closely related to *Hd2*. *Hd2* is a gene related to rice heading date regulation, which can inhibit heading date under long-day condition [52]. Therefore, we speculated that *SiCCT35* and *SiCCT11* might play an important role in regulating the heading time of foxtail millet. In addition to the genes in the *PRR* subfamily, some genes in the *COL* and *CMF* subfamilies show strong diurnal rhythms. These results suggested that many genes in the *SiCCT* family contributed an important function in the regulation of foxtail millet diurnal rhythm.

In addition to photoperiod regulation, the function of some *CCT* genes is also associated with abiotic stresses. Among such genes, many members of the *CMF* subfamily have been reported, such as *ZmCCT* and *GHD7* [22,24]. In the present study, many of the *CMF* subfamily genes showed sensitivity to ABA and stress treatment (Figure 8). In response to ABA treatment, all genes except *SiCCT3* and *SiCCT12* were up-regulated. Analysis of *cis*-acting elements (Figure 6) revealed that no ABA-responsive element was detected in *SiCCT3* and *SiCTT12*, which might be the reason why their expression was not induced by exogenous ABA. Most *CMF* family members were induced by drought stress except *SiCCT32*. They also had different degrees of response to low temperature and salt stress. Interestingly, *SiCCT37* expression was up-regulated in response to all treatments. Thus, this gene may have multiple functions, and the further study of the expression pattern and functions of this gene may be of importance for elucidating abiotic stress responses.

The foxtail millet *CCT* gene family was analyzed in this study. Thirty-seven full-length *SiCCT* genes were characterized and classified into three subfamilies according to their domains. The collinearity analysis and phylogenetic relationships of *SiCCT* genes provide valuable clues for dissecting the evolutionary characteristics of the *SiCCT* genes. The expression analysis of *SiCCT* genes responding to diurnal rhythm and abiotic stresses provide insights into their potential functions in the growth and development of foxtail millet. The present research will be helpful for further functional research into *SiCCT* genes and, potentially, for the improvement of the environmental adaptability and stress tolerance of foxtail millet.

## Figures and Tables

**Figure 1 genes-13-01829-f001:**
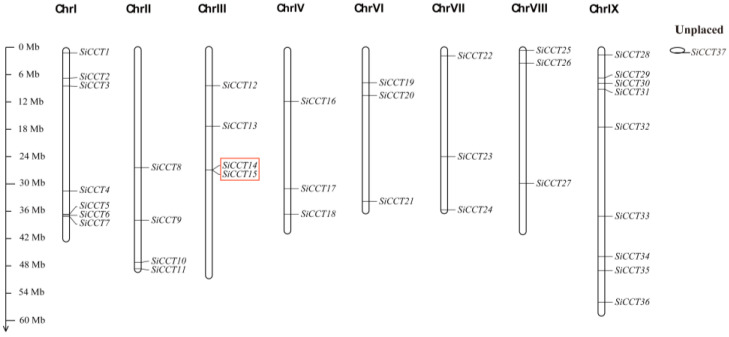
Chromosomal location of putative *SiCCT* genes. Thirty-six genes were mapped to the millet chromosomes, and one predicted gene was not mapped. The chromosome number is shown above each chromosome, and the length of each chromosome is indicated on the left in megabases (MB).

**Figure 2 genes-13-01829-f002:**
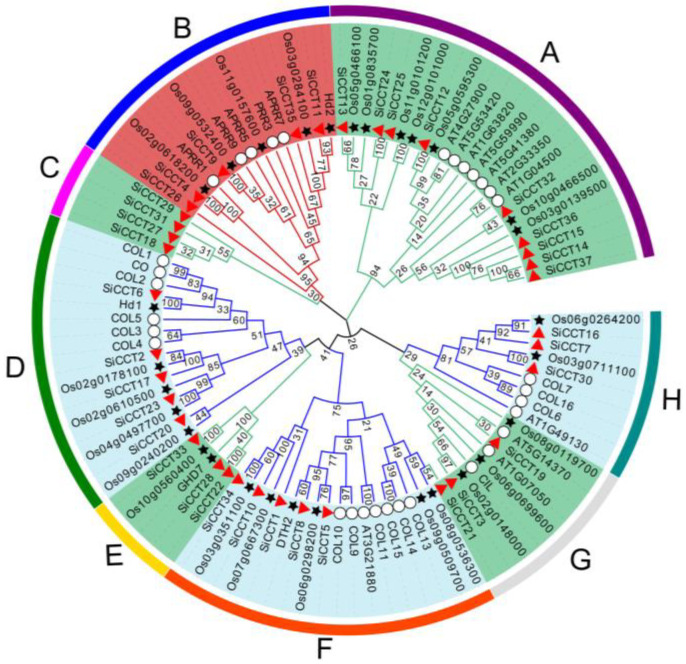
Phylogenetic relationships of CCT proteins. The phylogenetic tree was constructed from 100 protein sequences from millet, rice, and *Arabidopsis* using the maximum likelihood method with 1000 bootstrap replicates. The lines of different colors indicate different groups. Three subfamilies of CCT proteins are represented by different colored lines. Hollow circles, black stars, and red triangles represent the CCT proteins of Arabidopsis, rice, and millet, respectively.

**Figure 3 genes-13-01829-f003:**
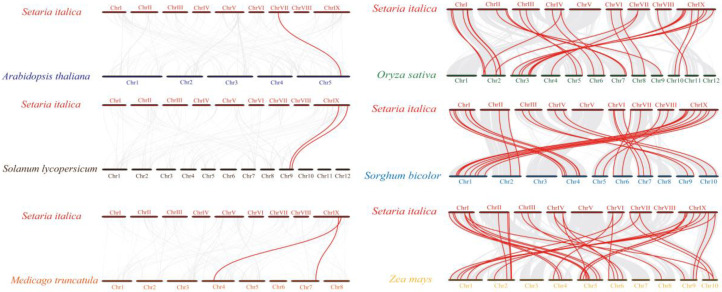
Homology of *CCT* genes between foxtail millet and six representative plant species. Gray lines in the background represent collinear blocks in the genomes of millet and the other plant species, and red lines highlight collinear *CCT* gene pairs.

**Figure 4 genes-13-01829-f004:**
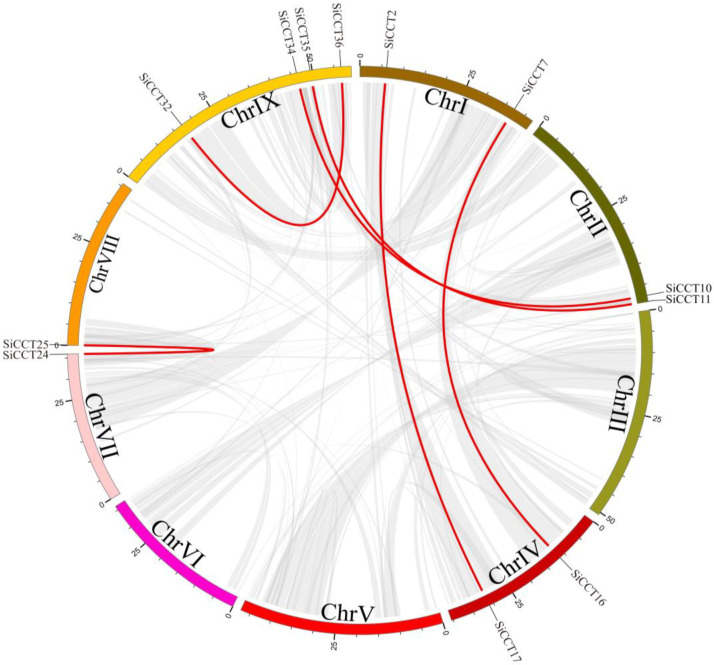
Interchromosomal relationships of millet *CCT* genes. Gray lines represent all common blocks in the millet genome, and red lines represent replicated *CCT* gene pairs.

**Figure 5 genes-13-01829-f005:**
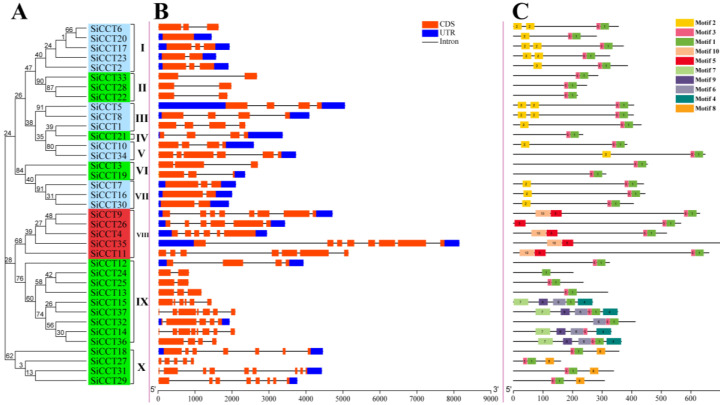
Phylogenetic relationships, gene structure, and conserved motifs of SiCCT proteins. (**A**) Phylogenetic tree constructed based on the domains of the foxtail millet CCT proteins. In the phylogenetic tree, blue areas are members of the *COL* family, green areas are members of the *CMF* family, and yellow areas are members of the PRR family. (**B**) Exon–intron structure of foxtail millet *CCT* genes. Blue boxes indicate the 5′ and 3′ untranslated regions; a red box represents an exon, and black lines represent introns. (**C**) Motif composition of foxtail millet CCT proteins. The motifs, numbered 1–10, are indicated by different colored boxes.

**Figure 6 genes-13-01829-f006:**
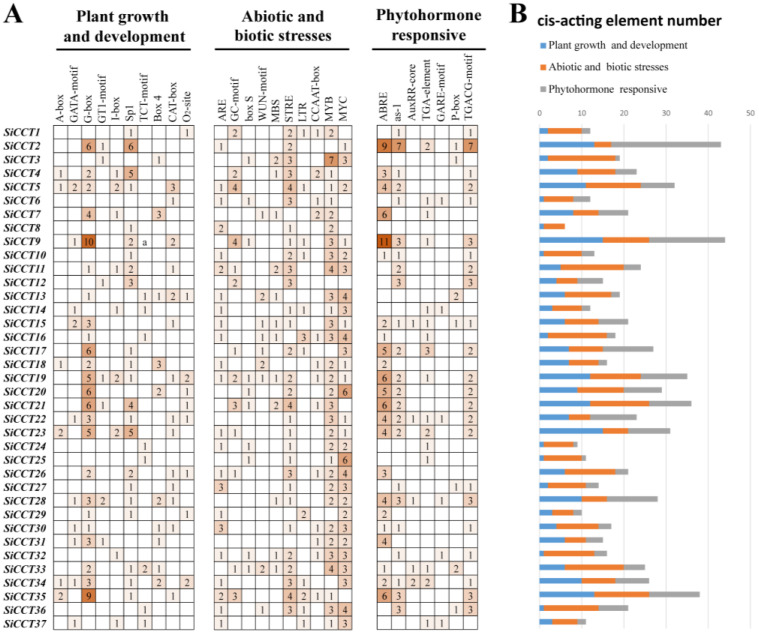
*Cis*-acting element analysis of the *CCT* gene family in foxtail millet. (**A**) Number of different promoter elements in the *CCT* genes indicated by different colors and numbers. (**B**) Number of *cis*-acting elements in three functional categories.

**Figure 7 genes-13-01829-f007:**
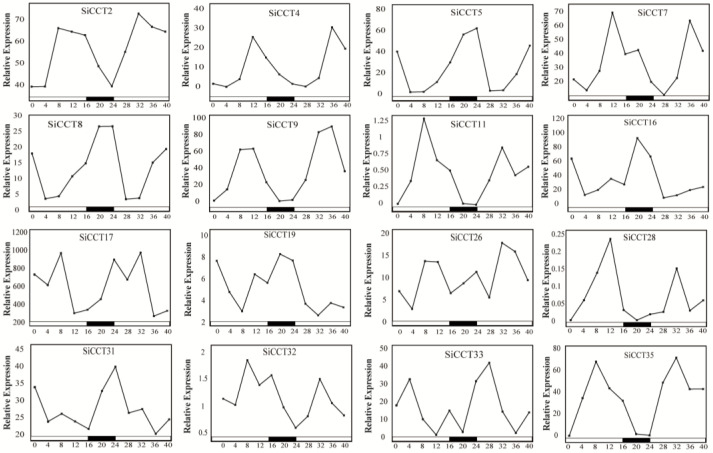
Expression pattern of SiCCT proteins under long days. White and black bars represent light and dark periods, respectively. The gene expression level was the FPKM value calculated.

**Figure 8 genes-13-01829-f008:**
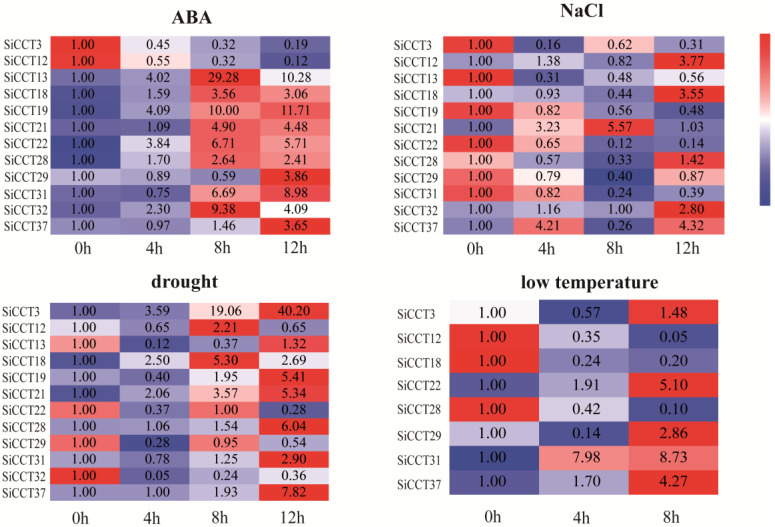
Expression profile of *SiCCT* genes under different treatments. Red boxes indicate up-regulation; blue boxes indicate down-regulation, and the intensity of the color indicates the intensity of expression. The color intensity of each color block is compared separately for each gene. The numbers on the graph were the expression levels of genes calculated by the 2^−ΔΔ*C*t^ method.

**Table 1 genes-13-01829-t001:** List of *SiCCT* genes identified in this study.

Gene Name	Transcript ID	Alias	Chr	Protein Length (aa)	MW (D)	pI	Location
*SETIT_019213mg*	KQL27760	*SiCCT1*	Ⅰ	432	46,243.27	5.60	Nucleus
*SETIT_017487mg*	KQL28473	*SiCCT2*	Ⅰ	386	39,459.15	6.32	Nucleus
*SETIT_017125mg*	KQL28750	*SiCCT3*	Ⅰ	453	47,725.85	8.14	Nucleus
*SETIT_016922mg*	KQL30521	*SiCCT4*	Ⅰ	518	57,741.59	5.59	Nucleus/Mitochondrion
*SETIT_017374mg*	KQL31286	*SiCCT5*	Ⅰ	407	44,610.57	4.99	Nucleus
*SETIT_019803mg*	KQL31326	*SiCCT6*	Ⅰ	355	39,793.63	8.00	Nucleus
*SETIT_017124mg*	KQL31374	*SiCCT7*	Ⅰ	440	47,328.18	6.54	Nucleus
*SETIT_030034mg*	KQL24119	*SiCCT8*	Ⅱ	406	43,630.40	5.20	Nucleus
*SETIT_029202mg*	KQL25444	*SiCCT9*	Ⅱ	630	70,287.20	6.22	Nucleus
*SETIT_030140mg*	KQL27073	*SiCCT10*	Ⅱ	384	40,363.07	6.10	Nucleus
*SETIT_033274mg*	KQL27318	*SiCCT11*	Ⅱ	661	71,442.89	8.74	Nucleus
*SETIT_022659mg*	KQL13967	*SiCCT12*	Ⅲ	325	35,522.48	4.93	Nucleus
*SETIT_024852mg*	KQL15112	*SiCCT13*	Ⅲ	319	35,116.48	5.75	Nucleus
*SETIT_024038mg*	KQL16011	*SiCCT14*	Ⅲ	331	36,436.76	4.58	Nucleus
*SETIT_024806mg*	KQL16012	*SiCCT15*	Ⅲ	268	29,552.18	4.78	Nucleus
*SETIT_006432mg*	KQL10228	*SiCCT16*	Ⅳ	445	47,590.03	5.09	Nucleus
*SETIT_006690mg*	KQL11132	*SiCCT17*	Ⅳ	372	39,574.38	6.13	Nucleus
*SETIT_006750mg*	KQL11783	*SiCCT18*	Ⅳ	357	37,898.91	5.01	Nucleus
*SETIT_014122mg*	KQL01011	*SiCCT19*	Ⅵ	313	33,098.76	7.65	Nucleus
*SETIT_014224mg*	KQL01176	*SiCCT20*	Ⅵ	281	28,425.62	6.42	Nucleus
*SETIT_014037mg*	KQL02690	*SiCCT21*	Ⅵ	235	26,147.14	9.58	Nucleus
*SETIT_011920mg*	KQK96183	*SiCCT22*	Ⅶ	217	23,959.73	6.75	Nucleus
*SETIT_010592mg*	KQK97912	*SiCCT23*	Ⅶ	326	34,393.37	5.15	Nucleus
*SETIT_011862mg*	KQL00040	*SiCCT24*	Ⅶ	202	21,587.85	5.61	Nucleus
*SETIT_027626mg*	KQK93230	*SiCCT25*	Ⅷ	236	25,887.70	6.50	Nucleus
*SETIT_026170mg*	KQK93667	*SiCCT26*	Ⅷ	566	62,003.23	7.40	Nucleus
*SETIT_027518mg*	KQK95022	*SiCCT27*	Ⅷ	160	17,635.30	10.18	Nucleus
*SETIT_039184mg*	KQK86275	*SiCCT28*	Ⅸ	248	27,292.51	6.86	Nucleus
*SETIT_036747mg*	KQK87242	*SiCCT29*	Ⅸ	308	32,998.37	8.66	Nucleus
*SETIT_035937mg*	KQK87465	*SiCCT30*	Ⅸ	406	44,451.22	6.65	Nucleus
*SETIT_036492mg*	KQK87704	*SiCCT31*	Ⅸ	339	35,998.10	4.77	Nucleus
*SETIT_035901mg*	KQK88797	*SiCCT32*	Ⅸ	412	43,116.93	4.37	Nucleus
*SETIT_036910mg*	KQK89929	*SiCCT33*	Ⅸ	286	29,324.74	7.14	Nucleus
*SETIT_034611mg*	KQK90887	*SiCCT34*	Ⅸ	648	69,184.42	8.49	Nucleus
*SETIT_034368mg*	KQK91381	*SiCCT35*	Ⅸ	760	82,668.80	6.07	Nucleus
*SETIT_039219mg*	KQK92647	*SiCCT36*	Ⅸ	365	39,857.30	4.67	Nucleus
*SETIT_020907mg*	KQK85264	*SiCCT37*	Unplaced	353	38,888.61	4.75	Nucleus

Chr: chromosome, aa: number of amino acids, MW: molecular weight, Da: Dalton, pI: theoretical isoelectric point.

**Table 2 genes-13-01829-t002:** Evolutionary analysis of *SiCCT* genes.

Type	Locus 1	Locus 2	*Ka*	*Ks*	*Ka*/*Ks*	T (MYA)	Period
Millet	*SiCCT32*	*SiCCT36*	1.04842	0.879676	1.19183	67.67	Intermediate ancestor
Millet	*SiCCT2*	*SiCCT17*	0.242598	1.71138	0.141756	131.64	Poaceae ancestor
Millet	*SiCCT7*	*SiCCT16*	0.218529	1.39626	0.15651	107.40	Poaceae ancestor
Millet	*SiCCT10*	*SiCCT34*	0.332322	2.14794	0.154716	165.23	Poaceae ancestor
Millet	*SiCCT24*	*SiCCT25*	0.00952499	0.0261684	0.363988	2.01	Millet
Millet	*SiCCT11*	*SiCCT35*	0.974077	1.10846	0.878766	85.27	Intermediate ancestor
Sorghum	*EES15491*	*EES07834*	0.0549948	0.12169	0.451925	9.36	Sorghum
Sorghum	*KXG22181*	*EES03844*	0.362499	2.54405	0.142489	195.70	Poaceae ancestor
Sorghum	*EES15491*	*EES03844*	0.879346	0.771143	1.14032	59.32	Intermediate ancestor
Sorghum	*KXG40217*	*EER94114*	0.952785	1.15926	0.821889	89.17	Intermediate ancestor
Sorghum	*EES05406*	*EES12446*	0.144635	1.20479	0.12005	92.68	Intermediate ancestor
Sorghum	*EES05581*	*EER88191*	0.997347	1.00659	0.990813	77.43	Intermediate ancestor
Sorghum	*OQU84486*	*EER90145*	1.01935	0.960586	1.06118	73.89	Intermediate ancestor
Sorghum	*EES07342*	*EER89633*	0.148272	0.900118	0.164726	69.24	Intermediate ancestor
Sorghum	*OQU85473*	*EER88227*	0.996734	1.01133	0.985566	77.79	Intermediate ancestor
Sorghum	*EER94860*	*EER99873*	0.347808	1.79793	0.19345	138.30	Poaceae ancestor
Maize	*Zm00001d025770*	*Zm00001d003162*	0.0261931	0.239901	0.109183	18.45	Intermediate ancestor
Maize	*Zm00001d021291*	*Zm00001d006212*	0.0625971	0.129561	0.483147	9.97	Maize
Maize	*Zm00001d022500*	*Zm00001d007107*	0.176657	0.333117	0.530314	25.62	Panicoideae ancestor
Maize	*Zm00001d029149*	*Zm00001d007107*	0.641455	0.939475	0.68278	72.27	Intermediate ancestor
Maize	*Zm00001d042958*	*Zm00001d012441*	0.0781679	0.396518	0.197136	30.50	Panicoideae ancestor
Maize	*Zm00001d013443*	*Zm00001d033719*	0.0617102	0.38183	0.161617	29.37	Panicoideae ancestor
Maize	*Zm00001d027598*	*Zm00001d014074*	1.01132	0.971378	1.04112	74.72	Intermediate ancestor
Maize	*Zm00001d032768*	*Zm00001d014074*	1.10514	0.795633	1.38901	61.20	Intermediate ancestor
Maize	*Zm00001d048369*	*Zm00001d014074*	0.5782	0.687273	0.841296	52.87	Intermediate ancestor
Maize	*Zm00001d036494*	*Zm00001d014656*	1.02009	0.946121	1.07818	72.78	Intermediate ancestor
Maize	*Zm00001d015468*	*Zm00001d046925*	0.510484	1.31613	0.387869	101.24	Poaceae ancestor
Maize	*Zm00001d025770*	*Zm00001d017176*	0.157993	1.59228	0.0992244	122.48	Poaceae ancestor
Maize	*Zm00001d051047*	*Zm00001d017176*	0.0802383	0.492435	0.162942	37.88	Panicoideae ancestor
Maize	*Zm00001d017241*	*Zm00001d051114*	0.0547288	0.209798	0.260864	16.14	Panicoideae ancestor
Maize	*Zm00001d051684*	*Zm00001d017885*	0.995396	1.01465	0.981028	78.05	Intermediate ancestor
Maize	*Zm00001d037327*	*Zm00001d017939*	0.857576	1.41978	0.604022	109.21	Poaceae ancestor
Maize	*Zm00001d045661*	*Zm00001d017939*	0.289121	1.6396	0.176336	126.12	Poaceae ancestor
Maize	*Zm00001d021291*	*Zm00001d052781*	1.01446	0.944687	1.07386	72.67	Intermediate ancestor
Maize	*Zm00001d025770*	*Zm00001d051047*	0.169024	1.43393	0.117874	110.30	Poaceae ancestor
Maize	*Zm00001d027598*	*Zm00001d048369*	1.007	0.96874	1.0395	74.52	Intermediate ancestor
Maize	*Zm00001d035134*	*Zm00001d049651*	0.985493	1.03346	0.953586	79.50	Intermediate ancestor
Maize	*Zm00001d045661*	*Zm00001d037327*	0.0757818	0.430623	0.175982	33.12	Poaceae ancestor
Maize	Zm00001d029149	*Zm00001d022500*	0.838386	0.587664	1.42664	45.20	Poaceae ancestor

*Ka*: non-synonymous rate, *Ks*: synonymous rate, T: divergent time, MYA: millions of years ago, T = *Ks*/(2 × 6.5 × 10^−9^) × 10^−6^ million years ago (Mya).

## Data Availability

The data and materials obtained in this study are available from the corresponding author on reasonable request.

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
