# Peer review of "Genome-Wide Identification and Characterization of the CCT Gene Family in Foxtail Millet (Setaria italica) Response to Diurnal Rhythm and Abiotic Stress"

_genes, 2022, doi:10.3390/genes13101829_

Round 1

Reviewer 1 Report

The authors of the article, "Genome-wide identification and characterization of the CCT gene family in foxtail millet (Setaria italica) under short days and abiotic stress clearly deliver information specifically on the title, they provide clear data on the CCT family in foxtail millet and present an interesting characterization at a genomic and phylogenetic scale (although I don't understand the definition of homology used in this article), but I consider they miss a great opportunity in presenting circadian rhythm expression data, especially on probable core oscillator components in this plant. The expression data need to be carefully revised; Figure 8 makes no sense to me, and there is no discussion on the tissue-specific data. I believe the article presents relevant and sound data, but there are details in the introduction, materials and methods, results, and discussion that need to be explained or revised.

The introduction of this article starts with flowering time, however, there are no experiments or discussion regarding this topic.

I was surprised that in an article describing CCT genes there are no circadian experiments, or at least take into account circadian regulation when presenting expression data (Fig. 7). PRR proteins are CCT-containing core circadian regulators. Most COL proteins are also circadian regulated, and there are no experiments aiming to study this aspect. 

I missed the discussion regarding tissue-specific expression, mainly because of particular expression profiles and sequence similarity to known flower, circadian, and stress regulators. 

There is some grammar and spelling mistakes noted in the document, please correct.

There are some particular observations that I refer to below:

Materials and methods

plant material: please provide more information on the cultivar. Millet germplasm has diversity regarding photoperiod sensitivity which might impact the results presented.

Information on qRT-PCR is missing. Which fluorophore was used? 

Melting curves and amplicon sizes would be very much appreciated.

I suppose (according to table S1) actin was used as a reference (calibrator) gene? 

DNase treatment was performed, or intron-spanning primers were designed?

I am a bit confused with the definition of homology by the authors. In Line 220 they describe homologous syntenic genes between different species, but in line 235 they describe homologous genes within foxtail millet. Are these genes, the product of possible genome duplications, truly homologous?

in section 3.4 the authors present a thorough cis-acting region analysis for all CCT genes and detect a G-box element involved in light in 24 of 37 genes. This element is clearly related to light and circadian rhythm expression, however, no light or circadian-dependent experiments are presented, which is a miss for me. 

In lines 67-68 (on page 14) the authors write: "In contrast, the expression of these genes 67 was maintained at a low level under illumination." Which genes? COL genes? Where are the results?

I don't understand figure 8; I don´t completely understand the color codes. According to the legend the values were calculated according to 2ddCt method, so I would believe the normalization value is 0h, therefore 1 in all first columns. But, it appears that their 0h values are not all the same.

In the ABA group, the values from 12h are strange. How come with value 10.28 there is no change, but with 2.41 there is?

This figure refers to drought experiments as "droughts". Please correct

There are also colors and numbers that do not make sense to me. 

Line 108 (page 16), CCT, please correct

Lines 160, page 17: the authors discuss tandem repeats in protein motifs within the family, but they do not refer to SiCCT29, 31, 27, and 18, which display their motifs in opposite ends; what about these proteins?

Lines 177-179 page 17: the authors write, "the COL family genes may have a similar function under SD and play an important role in regulating the heading date of foxtail millet". I would say this sentence could be enriched with information on the foxtail millet cultivar used; is it a short-day, neutral or long-day plant?

Lines 188-189: The authors write, "Approximately half of the genes were up-regulated under low temperature and salt stress." The authors have to be careful about this statement. They do not present paired controls, and previous reports have shown disruption of circadian control in response to abiotic stresses and time-of-day sensitivity. This is especially important if the genes characterized are circadian-regulated. 

Lines 193-194: The authors write, "The foxtail millet CCT gene family was comprehensively analyzed in this study." I completely disagree with this statement. No circadian rhythm studies were performed and the PRR homologs were ignored in this study. Please explain or correct.

I would also recommend relevant research regarding COL proteins:

Shalmani, A., Jing, X. Q., Shi, Y., Muhammad, I., Zhou, M. R., Wei, X. Y., ... & Chen, K. M. (2019). Characterization of B-BOX gene family and their expression profiles under hormonal, abiotic and metal stresses in Poaceae plants. BMC genomics, 20(1), 1-22.

Khanna, R., Kronmiller, B., Maszle, D. R., Coupland, G., Holm, M., Mizuno, T., & Wu, S. H. (2009). The Arabidopsis B-box zinc finger family. The Plant Cell, 21(11), 3416-3420.

Author Response

Dear Reviewer:

We would like to thank you for your comments and suggestions regarding our manuscript entitled “Genome-wide identification and characterization of the CCT gene family in foxtail millet (Setaria italica) under short days and abiotic stress” (Manuscript ID: genes-1915300). We have revised the structure and results of the article and revised the title to “Genome-wide identification and characterization of the CCT gene family in foxtail millet (Setaria italica) response to diurnal rhythm and abiotic stress”. The modified version was submitted. Meanwhile, the high-definition pictures in the “manual-supplementary.zip” were also uploaded. We have compiled the main corrections and responses to reviewers' comments below. We hope that the corrections will meet your approval.

Reviewer 1

The introduction of this article starts with flowering time, however, there are no experiments or discussion regarding this topic.

Response: Thanks for your helpful comment and suggestion. The flowering time content was removed from the introduction. Meanwhile the introduction had been modified lines 36 to 41, and the rhythm part was added in line 66 to 79 (on page 14) of modified version.

I was surprised that in an article describing CCT genes there are no circadian experiments, or at least take into account circadian regulation when presenting expression data (Fig. 7). PRR proteins are CCT-containing core circadian regulators. Most COL proteins are also circadian regulated, and there are no experiments aiming to study this aspect. 

 Response: Thank you very much for your suggestions. We had used transcriptome data to reanalyze the rhythms of genes in the CCT family and replaced Figure 7 with new results. The results and discussion were modified in line 66 to 79 (on page 14) and 209 to 218 (on page 18) of modified version.

I missed the discussion regarding tissue-specific expression, mainly because of particular expression profiles and sequence similarity to known flower, circadian, and stress regulators. 

 Response: Thank you very much for your comment. The tissue-specific expression was performed, which helped us to understand the expression characteristics of SiCCT genes. Fig. 7 had been changed to supplement Fig. S5

There is some grammar and spelling mistakes noted in the document, please correct.

  Response: Thank you very much for your suggestion. We had examined it carefully and corrected the mistakes in modified version.

There are some particular observations that I refer to below:

Materials and methods

plant material: please provide more information on the cultivar. Millet germplasm has diversity regarding photoperiod sensitivity which might impact the results presented.

 Response: Thank you very much for your suggestion. Yugu1 is a short-day variety. And it is a photoperiod-sensitive millet material, and long-day will delay the heading date.

Information on qRT-PCR is missing. Which fluorophore was used? 

 Response: Thank you very much for your suggestion. The GoTaq® qPCR Master Mix kit (Promega) was used for qRT-PCR reactions using a the C1000 Thermal Cycler and quantified using CFX96 Real-Time System (Bio-Rad, Hercules, CA, USA). Quantification of amplified products was done using the SYBR® Green I fluorophore.

Melting curves and amplicon sizes would be very much appreciated.

 Response: Thank you for your suggestion. We apologize for this omission. To examine the specificity of qRT-PCR rection, melting curves analysis was performed following the amplification. And the amplicon sizes used in RT-PCR and qRT-PCR were designed between 100 and 180 bp.

I suppose (according to table S1) actin was used as a reference (calibrator) gene? 

 Response: Thank you very much for your suggestion. We had added information about the Actin gene (SETIT_010361mg) in Table S1 and the manuscript in line 206.

DNase treatment was performed, or intron-spanning primers were designed?

  Response: Thank you very much for your comments. Total RNA was extracted from each tissue using a plant RNA extraction kit (Ac-curate Biotechnology Co., Ltd., Hunan, China), which included a DNase treatment step. The primers of reference gene were designed to span intron for monitoring DNA residues.

I am a bit confused with the definition of homology by the authors. In Line 220 they describe homologous syntenic genes between different species, but in line 235 they describe homologous genes within foxtail millet. Are these genes, the product of possible genome duplications, truly homologous?

  Response: Thank you very much for your comments. We had modified the description lines 270 to 273. We have modified the description to “Collinearity within the SiCCT family was analyzed to explore segmental duplication events. Six pairs of segmental-duplicated gene were found on foxtail millet chromosomes”.

in section 3.4 the authors present a thorough cis-acting region analysis for all CCT genes and detect a G-box element involved in light in 24 of 37 genes. This element is clearly related to light and circadian rhythm expression, however, no light or circadian-dependent experiments are presented, which is a miss for me. 

 Response: Thank you very much for your comments. We had added transcriptome analysis of rhythms and rewrote the results and discussions in line 66 to 79 (on page 14) and 209 to 218 (on page 18).

In lines 67-68 (on page 14) the authors write: "In contrast, the expression of these genes 67 was maintained at a low level under illumination." Which genes? COL genes? Where are the results?

 Response: Thank you very much for your comments. We had used transcriptome data to reanalyze the rhythms of genes in the CCT family and replace Figure 7 with new results and rewrote the results.

I don't understand figure 8; I don´t completely understand the color codes. According to the legend the values were calculated according to 2ddCt method, so I would believe the normalization value is 0h, therefore 1 in all first columns. But, it appears that their 0h values are not all the same.

  Response: Thank you very much for your comments. We had replaced Figure 7 with new results and rewrote the results.

In the ABA group, the values from 12h are strange. How come with value 10.28 there is no change, but with 2.41 there is?

 Response: Our expression analysis heat map is drawn separately based on the expression of a single gene, and the color indicates the extent to which a single gene is up or down. We were very sorry for the trouble caused to you. And we had added relevant instructions to the legend in lines 135 (on page 16).

This figure refers to drought experiments as "droughts". Please correct

 Response:  We apologize for this omission. We have modified the words in the picture.

There are also colors and numbers that do not make sense to me. 

 Response: Thank you very much for your comments. We will pay attention to the use of colors and numbers.

Line 108 (page 16), CCT, please correct

  Response: We apologize for this omission. We had modified the error in line 146 (on page 17).

Lines 160, page 17: the authors discuss tandem repeats in protein motifs within the family, but they do not refer to SiCCT29, 31, 27, and 18, which display their motifs in opposite ends; what about these proteins?

 Response: Thank you very much for your suggestion. We had added a description and discussion in line 24 to 26(on page 12) and 201 to 204 (on page 18) of modified version.

Lines 177-179 page 17: the authors write, "the COL family genes may have a similar function under SD and play an important role in regulating the heading date of foxtail millet". I would say this sentence could be enriched with information on the foxtail millet cultivar used; is it a short-day, neutral or long-day plant?

 Response: Thank you very much for your suggestion. Yugu1 is a short-day variety. And it is a photoperiod-sensitive millet material, and long-day will delay the heading date.

Lines 188-189: The authors write, "Approximately half of the genes were up-regulated under low temperature and salt stress." The authors have to be careful about this statement. They do not present paired controls, and previous reports have shown disruption of circadian control in response to abiotic stresses and time-of-day sensitivity. This is especially important if the genes characterized are circadian-regulated. 

 Response: Thank you very much for your suggestion. We had modified the description of gene expression under low temperature and salt treatment in line 248 (on page 18).

Lines 193-194: The authors write, "The foxtail millet CCT gene family was comprehensively analyzed in this study." I completely disagree with this statement. No circadian rhythm studies were performed and the PRR homologs were ignored in this study. Please explain or correct.

 Response: Thank you very much for your suggestion. We had added rhythm analysis of CCT family genes to the results using transcriptome data and analyzed these results in line 66 to 79 (on page 14). The sentence has been modified to “The foxtail millet CCT gene family was analyzed in this study.” in line 254 (on page 19) of modified version.

I would also recommend relevant research regarding COL proteins:

Shalmani, A., Jing, X. Q., Shi, Y., Muhammad, I., Zhou, M. R., Wei, X. Y., ... & Chen, K. M. (2019). Characterization of B-BOX gene family and their expression profiles under hormonal, abiotic and metal stresses in Poaceae plants. BMC genomics, 20(1), 1-22.

Khanna, R., Kronmiller, B., Maszle, D. R., Coupland, G., Holm, M., Mizuno, T., & Wu, S. H. (2009). The Arabidopsis B-box zinc finger family. The Plant Cell, 21(11), 3416-3420.

Response: Thank you very much for your suggestion. We had quoted these two references in the manuscript in lines 153.

Reviewer 2 Report

The article entitled “Genome-wide identification and characterization of the CCT gene family in foxtail millet (Setaria italica) under short days and abiotic stress” is a well-written manuscript, and the authors captured 37 putative SiCCT genes from the foxtail millet genome. Also, they studied the expression patterns under short days, abiotic stress and exogenous hormonal treatment. The expression of many CMF subfamily genes was significantly changed. Especially after drought treatment, expression of CMF subfamily genes except SiCCT32 was significantly up-regulated. In general, the results are innovative, significant, and useful for foxtail millet research. However, minor revisions should be incorporated into the manuscript.
1. Please italic the ‘Arabidopsis’ throughout your manuscript. Moreover, once you have mentioned scientific name of crop then no need to repeat further. For this check line 40 for Arabidopsis, repeated in line 57. Same for rice, its scientific name specified in line 52 but repeated in line 55.
2. Line 56, italic the scientific names of wheat and maize.

3. Line 124, correct the spell ‘rise’ to ‘rice’.
4. The stress of ABA, salt, drought, low temperature was given in greenhouse conditions whereas SD was given under field conditions. Whereas Line 159 read as ‘The roots, stems, leaves, seeds, and ears were collected separately at three developmental stages from field-grown millet plants for semiquantitative RT-PCR’. Kindly clarify the discrepancy.

5. Minor language editing is required for polishing of the manuscript. E.g primers used in this experiment are shown in Table S1 should be written as Primers used in this experiment are enlisted in Table S1.

6. Line 178 ‘The genes were designated the genes SiCCT1 to  SiCCT37’, correct as ‘from’

7. Line 181 change ‘were’ to ‘are’

8. Line 228 ‘may play’ to ‘may played’ be corrected.

9. In results it is mentioned “investigate the expression of the SiCCT genes in various tissues, comprising the leaf, immature seed, spikelet, root, leaf sheath, shoot apical meristem, stem, immature leaf, and seed (Figure 7)’ Lines 52-53 page 13. But in material and methods it is specified “The roots, stems, leaves, seeds, and ears were collected separately at three developmental stages from field-grown millet plants for semiquantitative RT-PCR” line 159 page 4. Clarify it.

10. Fig. 5 is mislabeled, phylogenetic tree should be labeled as ‘A’, gene structure as ‘B’ and motifs as ‘C’. Also, correct the citation of Fig 5A as 5B, 5B as 5C. Phylogenetic tree as Fig 5A is not cited in text.

11. Fig. 7 caption- correct SiCCT proteins to SiCCT genes. Also correct the caption of Fig. 9, it is not all hormone treatments.

12. Page 13, Line 56 correct ‘and two genes (SiCCT14 and SiCCT24) were not expressed in all tissues’ to and two genes (SiCCT14 and SiCCT16) were not expressed in all tissues.
13. Highlight the novelty of your findings and it would be more interesting if the authors focus more on the significance of their findings.

Author Response

Dear Reviewer:

We would like to thank you for your comments and suggestions regarding our manuscript entitled “Genome-wide identification and characterization of the CCT gene family in foxtail millet (Setaria italica) under short days and abiotic stress” (Manuscript ID: genes-1915300). We have revised the structure and results of the article and revised the title to “Genome-wide identification and characterization of the CCT gene family in foxtail millet (Setaria italica) response to diurnal rhythm and abiotic stress”. The modified version was submitted. Meanwhile, the high-definition pictures in the “manual-supplementary.zip” were also uploaded. We have compiled the main corrections and responses to reviewers' comments below. We hope that the corrections will meet your approval.

Reviewer 2

  1. Please italic the ‘Arabidopsis’ throughout your manuscript. Moreover, once you have mentioned scientific name of crop then no need to repeat further. For this check line 40 for Arabidopsis, repeated in line 57. Same for rice, its scientific name specified in line 52 but repeated in line 55.

Response: We apologize for this omission. We had carefully examined the manuscript and modified the unitalic 'Arabidopsis' in it, and carefully examined the use of botanical names and changed the error.

  1. Line 56, italic the scientific names of wheat and maize.

Response: Thank you very much for your suggestion. We had corrected the mistake in line 67.

  1. Line 124, correct the spell ‘rise’ to ‘rice’.

Response: We apologize for this omission. We had corrected the error of the wrong word in lines 153.

  1. The stress of ABA, salt, drought, low temperature was given in greenhouse conditions whereas SD was given under field conditions. Whereas Line 159 read as ‘The roots, stems, leaves, seeds, and ears were collected separately at three developmental stages from field-grown millet plants for semiquantitative RT-PCR’. Kindly clarify the discrepancy.

Response: Thank you very much for your suggestion. We had modified the information regarding the growth conditions of the study materials in line 188.

  1. Minor language editing is required for polishing of the manuscript. E.g primers used in this experiment are shown in Table S1 should be written as Primers used in this experiment are enlisted in Table S1.

Response: Thank you very much for your suggestion. We had revised this description in line 205 and examined the entire manuscript.

  1. Line 178 ‘The genes were designated the genes SiCCT1 to SiCCT37’, correct as ‘from’

Response: We apologize for this omission. We had modified this sentence in line 215.

  1. Line 181 change ‘were’ to ‘are’

Response: We apologize for this omission. We had modified this tense problem in line 218.

  1. Line 228 ‘may play’ to ‘may played’ be corrected.

Response: We apologize for this omission. This problem had been corrected in line 265.

  1. In results it is mentioned “investigate the expression of the SiCCT genes in various tissues, comprising the leaf, immature seed, spikelet, root, leaf sheath, shoot apical meristem, stem, immature leaf, and seed (Figure 7)’ Lines 52-53 page 13. But in material and methods it is specified “The roots, stems, leaves, seeds, and ears were collected separately at three developmental stages from field-grown millet plants for semiquantitative RT-PCR” line 159 page 4. Clarify it.

Response: Thank you very much for your suggestion. We had changed the description of the problem in the material method lines 188 to 189.

  1. Fig. 5 is mislabeled, phylogenetic tree should be labeled as ‘A’, gene structure as ‘B’ and motifs as ‘C’. Also, correct the citation of Fig 5A as 5B, 5B as 5C. Phylogenetic tree as Fig 5A is not cited in text.

Response: We apologize for this omission. We had made changes to figure 5 and corrected the citation of Fig 5A as 5B, 5B as 5C. Andwe added the description of figure 5A in line 2 (on page 12).

  1. Fig. 7 caption- correct SiCCT proteins to SiCCT genes. Also correct the caption of Fig. 9, it is not all hormone treatments.

Response: We apologize for this omission. We had corrected these the mistake in line 133 (on page 16).

Fig. 7 had been changed to supplement Fig. S5.

  1. Page 13, Line 56 correct ‘and two genes (SiCCT14 and SiCCT24) were not expressed in all tissues’ to and two genes (SiCCT14 and SiCCT16) were not expressed in all tissues.

Response: We apologize for this omission. We had changed ‘(SiCCT14 and SiCCT24)'to ‘(SiCCT14 and SiCCT16)’ in line 60 (on page 13).

  1. Highlight the novelty of your findings and it would be more interesting if the authors focus more on the significance of their findings.

Response: Thank you very much for your comments.

Round 2

Reviewer 1 Report

This second version is an improvement of the first. There are some minor english errors that should be corrected.

It is a good idea to use another gene expression set for diurnal analysis. The authors refer to the analysis shown in figure 7 as diurnal rhythms, but in lines 64 - 77 (65-78 in version with corrections) there are some mentions of circadian rhythms of the PRRs. The data set present do not show circadian rhythms which should be maintained under continuous (light or dark) conditions. I understood from the cited article that samples are taken during 24 h, but from plants under a light/dark cycle, hence diurnal. Please correct.

Author Response

Dear Reviewer: We would like to thank you for your letter and the reviewers’ comments and suggestions regarding our manuscript entitled “Genome-wide identification and characterization of the CCT gene family in foxtail millet (Setaria italica) response to diurnal rhythm and abiotic stress” (Manuscript ID: genes-1915300). We have modified the manuscript and uploaded a modified version with a tag. We have compiled the main corrections and responses to reviewers' comments below. We hope that the corrections will meet your approval. Reviewer It is a good idea to use another gene expression set for diurnal analysis. The authors refer to the analysis shown in figure 7 as diurnal rhythms, but in lines 64 - 77 (65-78 in version with corrections) there are some mentions of circadian rhythms of the PRRs. The data set present do not show circadian rhythms which should be maintained under continuous (light or dark) conditions. I understood from the cited article that samples are taken during 24 h, but from plants under a light/dark cycle, hence diurnal. Please correct. Response:Thank you very much for your suggestion. We had changed ‘circadian rhythm’ to ‘diurnal rhythm’, and we also modified other errors including wrong sentences and words in the manuscript. At the same time, we joined the blue and red lines in Figure 7. The blue line showed sampling at the beginning of dawn, and the red line showed sampling at the beginning of darkness. The sampling overlapped between 16-24 h. Therefore, we used of the average values of two sets of data to draw the line graph during 16-24 h in the Figure 7.
